# Scalable and accurate method for neuronal ensemble detection in spiking neural networks

Rubén Herzog[1]*, Arturo Morales[2], Soraya Mora[3,4], Joaquín Araya[1,5], María-José Escobar[2], Adrian G. Palacios[1], Rodrigo Cofré[6]

1 Centro Interdisciplinario de Neurociencia de Valparaíso, Universidad de Valparaíso, Valparaíso, Chile, 2 Departamento de Electrónica, Universidad Técnica Federico Santa María, Valparaíso, Chile, 3 Facultad de Medicina y Ciencia, Universidad San Sebastián, Santiago, Chile, 4 Laboratorio de Biología Computacional, Fundación Ciencia y Vida, Santiago, Chile, 5 Escuela de Tecnología Médica, Facultad de Salud, Universidad Santo Tomás, Santiago, Chile, 6 CIMFAV Ingemat, Facultad de Ingeniería, Universidad de Valparaíso, Valparaíso, Chile

* rubenherzog@ug.uchile.cl

**Data Availability Statement:** The GUI, codes, and data required for the use and validation of our method is available on GitHub (https://github.com/brincolab/NeuralEnsembles).

## Abstract

We propose a novel, scalable, and accurate method for detecting neuronal ensembles from a population of spiking neurons. Our approach offers a simple yet powerful tool to study ensemble activity. It relies on clustering synchronous population activity (population vectors), allows the participation of neurons in different ensembles, has few parameters to tune and is computationally efficient. To validate the performance and generality of our method, we generated synthetic data, where we found that our method accurately detects neuronal ensembles for a wide range of simulation parameters. We found that our method outperforms current alternative methodologies. We used spike trains of retinal ganglion cells obtained from multi-electrode array recordings under a simple ON-OFF light stimulus to test our method. We found a consistent stimuli-evoked ensemble activity intermingled with spontaneously active ensembles and irregular activity. Our results suggest that the early visual system activity could be organized in distinguishable functional ensembles. We provide a Graphic User Interface, which facilitates the use of our method by the scientific community.

## Introduction

Donald Hebb predicted more than 70 years ago that ensembles would naturally arise from synaptic learning rules, where neurons that fire together would wire together [1]. However, despite the long history of this idea, only recently the simultaneous recordings and computational analysis from hundreds of cells have turned out to be possible [2]. Recent advances in recording technology of neuronal activity combined with sophisticated methods of data analysis have revealed significant synchronous activity between neurons at several spatial and temporal scales [3–5]. Theses groups of neurons that have the tendency to fire together known as *neuronal ensembles* (also called cell assemblies) are hypothesized to be a fundamental unit of neural processes and form the basis of coherent collective brain dynamics [1, 6–8].

**Funding:** RH is funded by CONICYT scholarship CONICYT-PFCHA/Doctorado Nacional/2018-21180428. (https://www.anid.cl) AM is funded by CONICYT scholarship CONICYT-PFCHA/Magíster Nacional/2020- 22200156. (https://www.anid.cl) RC is funded by Fondecyt Iniciación 2018 Proyecto 11181072. (https://www.anid.cl) MJE and AGP are funded by AFOSR Grant FA9550-19-1-0002 (https://www.afrl.af.mil/AFOSR/). AGP is funded by ICM-ANID #P09-022-F, CINV (http://www.iniciativamilenio.cl/). SM and JA received no funding for developing this work. All the funders had no role in study design, data collection and analysis, decision to publish, or preparation of the manuscript.

**Competing interests:** The authors have declared that no competing interests exist.

The idea that the functional units of neural processes are groups of neurons firing together (not single neurons) represented a paradigm shift in the field of computational neuroscience [6]. Neuronal ensembles have been proposed as a fundamental building block of the whole-brain dynamics, and relevant to cognitive functions, in particular, as ensemble activity could implement brain-wide functional integration and segregation [3].

Large-scale neuronal recordings techniques such as multi-electrode arrays (MEA) or calcium imaging, allow for the recording of the activity of hundreds and even thousands of neurons simultaneously [5, 9–12]. These recent technological advances provide a fertile ground for analysing neuronal ensembles and investigating how collective neuronal activity is generated in the brain. Recent studies using multi-neuronal recording techniques have revealed that a hallmark of population activity is the organization of neurons into ensembles, generating new insights and ideas about the neural code [9, 13–15]. In particular, the activation of specific ensembles has been shown to correlate with spontaneous and stimuli evoked brain function [16]. The brain-wide alterations present in neurological and mental impairments disrupt neural population activity and therefore affect the neuronal ensembles. Indeed, neuronal ensembles are susceptible to epileptic seizures and schizophrenia as shown in *in vivo* two-photon calcium imaging data in mouse [17, 18], in medically-induced loss of consciousness in mice and human subjects [19] and in a mice model of autism [20].

However, identifying and extracting features of ensembles from high-dimensional spiking data is challenging. Neuronal ensembles have different sizes and have different activity rates. Some neurons may not participate in ensemble activity, while others may participate in many, and not all neurons within an ensemble fire when the ensemble is active. Ensembles can exhibit temporal extension, overlap, or display a hierarchical or modular organization, making it difficult to distinguish between them [2].

In fact, there is a wide variety of techniques and ideas that have been used to detect and interpret neuronal ensembles (see [13, 21, 22] for reviews), for example, previous works have applied different methodologies such as principal component analysis [23–26], correlation between neurons [26–28], correlation between population vectors [16], statistical evaluation of patterns [29–32] and non-negative matrix factorization [25, 33]. However, most of these methods are computationally expensive, and require tuning several parameters, which hinder their application by the scientific community.

These divergences on the definition of neural ensembles have been addressed by Carrillo and Yuste [34], where they define quantitatively neural ensembles as population vectors whose dimension is the size of the population. Here, we follow their definition and consider as population vector the binary response of the population on a given time bin. Also, we add the specification that a neural ensemble should have a core, a group of neurons whose temporal activity is significantly correlated with the temporal activity of the ensemble. On the opposite, non-core cells are neurons that can participate in the ensemble or not, but their correlation with the activation times of the ensemble is not statistically significant. Further, the core of an ensemble is the group of neurons that best represent the activity of the ensemble, so their presence indicates a functional coupling between ensembles and neurons. Note that, in general, the temporal activity of the ensemble cannot be determined by the sole activity of its core-cells, rather, it is a functional whole built up from core-cells and the recruitment of non-core cells.

Accordingly, we develop a method based on dimensionality reduction, clustering, and non-parametric statistical tests to detect neural ensembles on simultaneous neuronal recordings. We use Principal Component Analysis to project population vectors on a low-dimensional space where they can be reliably clustered. Once clustered, a non-parametric statistical test is performed to detect the core-cells, fulfilling our definition of ensembles as multi-dimensional population vectors with a core.

Moreover, we tested and validated our method using biological and simulated data, showing its accuracy and broad applicability to different scenarios recreated by synthetic data with diverse characteristics. Our tests on simulated data shows a remarkable detection performance for the ensemble number, the ensemble activation times, and the core-cells detection over a wide range of parameters.

Besides, we exemplify our method's functionality on spiking neuronal data recorded using MEA from mouse *in vitro* retinal patch, from which the spiking activity of hundreds of retinal ganglion cells (RGCs) is obtained during a simple ON-OFF stimuli, i.e. consecutive changes between light and darkness. In brief, the vertebrate retina is part of the central nervous system composed of thousand of neurons of several types [35–37], organized in a stratified way with nuclear and plexiform layers [38]. This neural network has the capability to process several features of the visual scene [39], whose result is conveyed to the brain through the optic nerve, a neural tract composed mainly by the axons of the RGCs. In fact, the physiological mechanisms involved in many of those processes are starting to become clear with the development of new experimental and computational methods [37, 38, 40]. One of the most remarkable, and simple, example of retinal processing is the ON-OFF responses of RGCs, a stereotypical increase or decrease in firing rate responding to changes light intensity. In this case, the connectivity between RGCs and Bipolar cells (Bc) in the Inner Plexiform Layer (IPL) plays a major role, determining the tendency of RGC to preferentially fire when the light increased, decreased, or both [38]; this property is often called polarity [41], and represents the broadest functional classification of RGCs into ON, OFF, and ON-OFF cell types.

We hypothesize that retinal ensembles may also exhibit this property as a whole. Our analysis suggests the existence of diverse ON and OFF retinal ensembles with a specific stimulus preference as functional units, which may implicate that a stimulus tuning preference is a property of the ensembles as a whole, and not a simple inheritance from their corresponding core-cells.

To facilitate our method's use by the community, we provide a Graphic User Interface and the codes that implement our algorithm that aim to provide a fast, scalable, and accurate solution to the problem of detecting neuronal ensembles in multi-unit recordings.

## Materials and methods

### Ethics statement

Regarding retinal data, animal manipulation and breeding and corresponding experiments were approved by the bioethics committee of the Universidad de Valparaiso, in accordance with the bioethics regulation of the National Agency for Research (ANID, Ex-CONICYT) and international protocols.

### Ensemble detection

We consider binary spike trains arranged in a matrix $\mathbf{S}_{N \times T}$ where $N$ corresponds to the number of neurons and $T$ the number of time bins. The entries of the matrix denoted by $\mathbf{S}_{n,t}$ are equal to one if the $n$-th neuron is active on the $t$-th time bin, and zero otherwise. At each time bin $t$, there is a binary population vector of active and silent neurons ($\mathbf{S}_{1,t}, \ldots, \mathbf{S}_{N,t}$).

**Feature extraction using PCA.** To reduce the dimensionality of the population vectors, we used Principal Component Analysis (PCA). PCA extracts a set of orthogonal directions capturing the most significant variability observed on the population vectors. We discarded population vectors with less than three spikes (this is a free parameters in the GUI) and projected the selected population vectors on the first six principal components (PCs). However, the optimal number of PCs may vary depending on the data, so it is also considered a free

parameter in the GUI. Its selection is informed by the cumulative explained variance of those PCs and by the cut-off of the eigenvalue spectrum of the PC decomposition. This plot is available in the GUI.

Finally, we computed the Euclidean distance between the population vectors projected on the PC space to obtain a distance matrix. The distance matrix characterizes the dissimilarity between population vectors and its analysis is essential in the clustering step.

**Centroid detection.** We used a hard clustering algorithm based in the deterministic analysis of the distance matrix. Two parameters characterized each data point (i.e., a population vector) in the clustering procedure: i) $\rho$ the density and ii) $\delta$ the minimum distance to a point with higher density. Conceptually, the density of a point represent how close (in terms of the distance matrix) the closest neighbours are. The density of each vector $i$ was given by $\rho_i = 1/\overline{d_{i,v}}$, where $\overline{d_{i,v}}$ was the average distance from point $i$ to its closest neighbours. Typically, we considered 2% of the closest neighbours. This choice can be tuned (in the GUI) with a parameter denoted by $d_c$. population vectors with relatively high values of $\rho$ and $\delta$ were candidates of centroids. The rationale is that points with relatively high $\rho$ and $\delta$ (respect to the rest of the points) have the highest number of points in their neighbourhood and are far from other points with high density. The procedure to find the centroids of the clusters follows the procedure presented in [42], which is a modified version of the method developed in [43]. To automatically detect the cluster centroids, we fit a power-law to the $\delta$ vs. $\rho$ curve, using the 99.9% upper confidence boundary as a threshold (another free parameter in the GUI). We considered centroids to be all points falling outside this boundary. The rest of the points were assigned to their closest centroids, building up the clusters. Since each population vector is associated to a time bin, the clustering step yields the cluster activation times.

**Core-cell detection.** Core-cells are the best representatives of a neural ensemble and their presence, measured by a statistical test, suggest coupling between the single neuron scale and the ensemble scale, i.e. multi-scale statistical dependence. Accordingly, once the candidate clusters and centroids were identified, the Pearson correlation coefficient between the activation times of neuron $n$ and cluster $e$, denoted by $corr(n, e)$, was computed for each pair $(n, e)$. This correlation is 1 if neuron $n$ and cluster $e$ were always active in the same time bins, and is -1 if they are never active at the same time bins. The intermediate values represented combinations of neurons and clusters with a tendency to be active either in the same or in different time bins.

To distinguish between a core and a non-core cell, we set a threshold obtained from a null hypothesis built from shuffled versions of the $(n, e)$ pair. For each cluster, $e \in E$, the number of times in which it was active is kept fixed, but the temporal activation was randomly shuffled, obtaining a new temporal sequence of activation that we denoted $e_r$.

We repeated this procedure 5000 times for each cluster, obtaining a distribution of $corr(n, e_r)$ for neuron $n$ in cluster $e$, which represented the null hypothesis distribution associated with the correlation between a neuron and a cluster given by chance. A significance threshold was then defined as the 99.9-th percentile of this distribution (free parameter); therefore, we set $p < 0.001$ as threshold $\theta_e$. Then, if $corr(n, e) > \theta_e$, we considered that $n$ was a core-cell of cluster $e$, as their correlation was above the significance level.

**Within-cluster average correlation.** To evaluate the inner structure of the cluster, we computed the within-cluster correlation by averaging the pairwise correlations of all core-cells of a given cluster. As a comparison, we also computed the average pairwise correlation of the whole population to obtain a threshold for ensemble selection.

**Ensemble selection criteria.** Following [44], we used two criteria: i) minimum cluster size and ii) within-cluster average correlation. The first criterion considered as candidates for neuronal ensembles clusters with a minimum number of core-cells. This ensures that the detected

cluster matches the definition of neural ensemble as a cluster of population vectors with a core. In this article we used three as the minimum number of core-cells, but this number is presented as a free parameter in the GUI and codes. The second criterion compares the within-cluster average correlation with the average pairwise correlation of the population. In this study, if the latter exceeds the former, we discard that cluster. Note that this threshold is also free parameters in the GUI and it is expressed in terms of the standard deviation of the pairwise correlations of the whole population.

This way, we kept only clusters with a minimum size (in terms of core-cells) and with relatively (respect to the population) high within-cluster average correlation. If a cluster failed to pass any of these two criteria, we discarded it from the analysis. Otherwise, we considered the cluster as an ensemble.

## Results

### Method overview

Before a systematic evaluation of the performance of our method, we summarize its core elements using the example shown in Fig 1, and refer the reader to the Methods section for further details.

First, we discard any population vector (Fig 1A) with less than three active neurons (a free parameters in the GUI). This way we focus only on high-order interactions, leaving pairwise interaction out of the analysis. Then, we perform a principal component analysis (PCA) using the selected population vectors as observations (Fig 1B). Then, we computed the Euclidean distance between all the vectors projected on the first six principal components (see Methods for details on the selection of the number of PCs). Once the pairwise distances between population vectors are computed, the local density, $\rho$, and the distance to the next denser point, $\delta$, are computed for each population vector. Based on these two measures, and a power-law fit to the $\rho$ vs. $\delta$ curve, we automatically detect the cluster centroids (Fig 1C), and assign the rest of the vectors to their closest centroid, building up the clusters (Fig 1D). Since the core of our algorithm is the density-based clustering procedure, we call it *Density-based*.

To find the core-cells, we computed $corr(n, e)$, the correlation between the spike train of neuron *n* and the activation times of cluster *e* (Fig 1E), and test its significance using a threshold associated with a null hypothesis obtained from shuffled versions of data (Fig 1F). If $corr(n, e)$ is above the threshold, neuron *n* is considered a core-cell of cluster *e* (Fig 1G).

Finally, to obtain the ensembles, the pairwise correlation between all the core-cells of a given cluster are computed and averaged, representing the within-cluster average correlation.

To define a cluster be an ensemble, we compared the within-cluster average correlation to the average pairwise correlation of the whole network (Fig 1H). If the within-cluster if significantly higher (based on a threshold), we considered an *ensemble* to be present; otherwise, the cluster is discarded due to the lack of internal correlation.

With this procedure, we were able to split the population vectors into ensemble and non-ensemble vectors (Fig 1I), and to obtain the activation times of different ensembles in time (Fig 1J) with their corresponding core-cells. We clarify that our method does not provide any analytical tool to evaluate the statistical significance of the ensemble temporal sequence, rather, it generates it as an integer sequence which we refer as global sequence (Fig 1J).

**Synthetic spike trains.** We generated a set of ensembles *E* characterized by their *core-cells* and an *activation sequence* of ensembles. Each ensemble was composed of a fixed number of core cells randomly drawn from the whole population of neurons, allowing the repetition of core-cells among ensembles. We generated for each ensemble, a column binary vector $c_e$ of dimension *N*, with $c_e(n) = 1$ if neuron *n* is a core-cell of ensemble *e* and 0 otherwise.

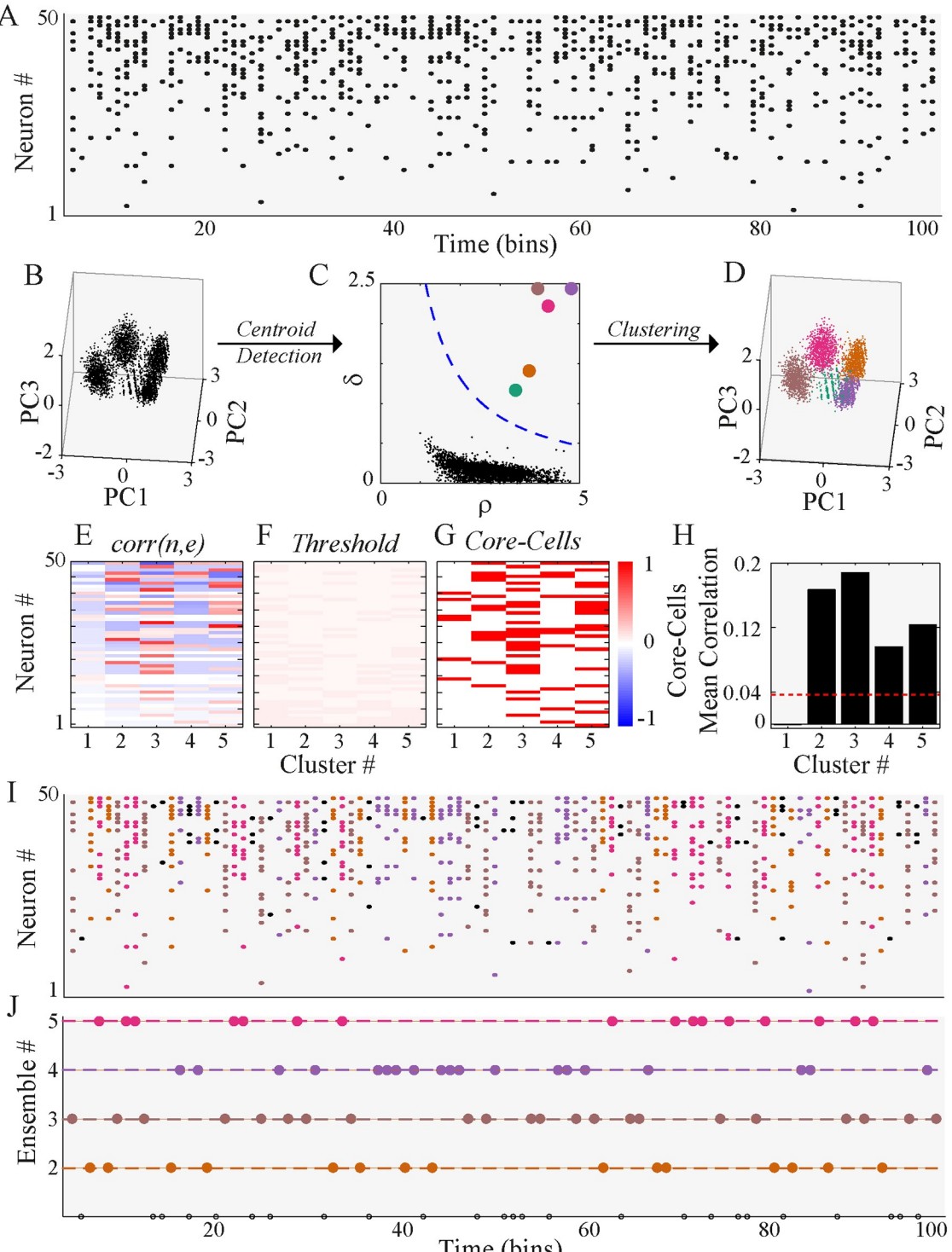

**Fig 1. Methodological scheme.** **(A)** Synthetic spike trains of 50 neurons during 100-time bins. Four ensembles were artificially generated (see Methods for details). **(B)** PCA is performed on all the population vectors of more than two active neurons, we used in this case three. For visualization purposes, we plot each of them as a point in the 3D space spanned by the first three principal components (PC). The points tend to group in clusters. **(C)** The Euclidean distance between all the points on the PC coordinates is computed. The density of each data point $\rho$ and its distance to the next data point with a higher density of $\delta$ is obtained. The points above the threshold (blue dashed line, see Methods for details) are considered cluster centroids. **(D)** All points are assigned to their closest centroid, building up the clusters of population vectors. **(E)** The correlation matrix between neurons and clusters, $corr(n, e)$ is represented by the symmetric red-blue color map. **(F)** A threshold matrix is computed to define the significance of $corr(n, e)$ values.

**(G)** Core-cells are detected based on $corr(n, e)$ values that exceed the threshold (red if core, white otherwise). **(H)** The inner-cluster mean correlation is compared against a threshold of non-core cell correlation, discarding any cluster below this threshold (cluster 1, in this case). This final filtering yields the set of ensembles. **(I)** Each population vector shown in **A** is coloured according to the cluster to which they belong. Black vectors were discarded by the minimum number of spikes or the threshold rejection criteria. **(J)** The detected ensemble sequence is represented as an integer sequence, where the color corresponds to the population vector color in **I**. Note that the rejection criteria discarded cluster 1, so only ensembles 2–5 are shown, and black circles denote non-clustered population vectors.

The activation sequence of each ensemble $e \in E$ follows a time homogeneous Bernoulli process with parameter $p_e$, denoted $B(p_e)$. We generate a row binary vector denoted by $a_e$ of dimension $T$ with $a_e(t) = 1$ if ensemble $e$ is active on time bin $t$ and 0 otherwise. We allowed at most one active ensemble per bin, so each population vector can be part of one ensemble or none. To implement this idea, we first drew the first ensemble's activation times at random and then removed these times from the list of possible time points available for the second ensemble. We proceeded this way until reaching $P_E$.

With the core-cells and the activation sequence of each ensemble defined, a spike train was generated by the product $c_e\, a_e$ (matrix) of dimension $N \times T$.

$$\mathbf{S}^E(n, t) = \sum_{e \in E} c_e(n) a_e(t).$$

In order to preserve the probabilistic nature of spiking neurons, the firing rates of each neuron was drawn from a rectified Gaussian distribution (only positive values) with variable standard deviation (s.d). The larger the s.d., the higher the firing rates are present on the spike train. This parameter allowed for the control of the density of the spike train (the total number of spikes respect to the spike train duration).

We randomly added/removed spikes to/from each neuron's spike train until matched the target firing rate $P(n)$. In removing spikes, the population vector located on the time bins where a given ensemble was active, end up having less active neurons than defined; on the other hand, if we added spikes, the opposite effect occurred. We used three different spike train densities: low (s.d. = 0.05), medium (s.d. = 0.1) and high (s.d. = 0.2). These densities reproduce a wide range of spiking responses observed experimentally, however, the density in experimental data will depend on the bin size. Our retinal data corresponds to the medium case. For the low-density case, matching the target firing rates usually required the removal of spikes. This method corrupted the vectors related to ensemble activity by under-representing their core-cells. In the medium case, we had a balance between adding and removing spikes, and, in the last case, most neurons required the addition of spikes to match the desired firing rate, making neurons participate in ensembles they did not belong by construction.

In summary, we generated a spike train from the following procedure:

1. Generate $E$ ensembles characterized by population vectors built from their core-cells.

2. Fill a spike train with the activation times of ensembles following a time-homogeneous Bernoulli process for each ensemble considering the proportion of $P_E$ of the complete set of population vectors. The remaining population vectors are filled with the population vector consisting only of silent neurons.

3. Draw the firing rates of each neuron from a rectified positive Gaussian distribution.

4. Randomly add/remove spikes to/from each neuron's spike train until it matches the target firing rate of each neuron.

This procedure yields a random spike train built from the ensemble activity.

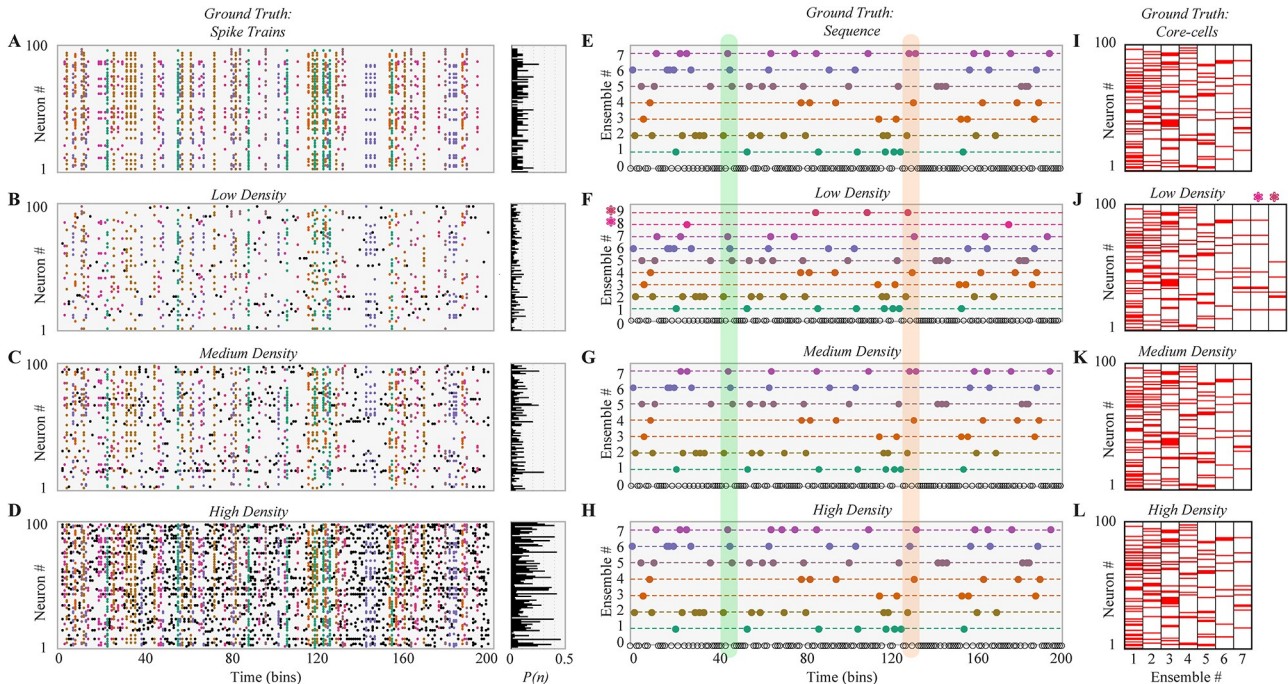

**Fig 2. Robust ensemble detection on synthetic data under different density regimes. (A)** The spatial (core-cells) and temporal (activation times) properties of seven ensembles are synthetically generated on a spike train of 100 neurons. The right panel shows the firing probability of each neuron at each time bin denoted by $P(n)$. Each ensemble is coloured differently. **(B-D)** Based on the ground truth (GT) spike trains, three spike trains with different densities were generated: low **B**, medium **C** and high density **D**. The firing probability of each neuron is shown at the right as in **A**. population vectors are coloured as in **A**. Black dots are population vectors that do not belong to any ensemble. **(E)** Activation times (i.e. sequence) of the GT ensembles. Colour corresponds with **A**. **(F-H)** The detected ensemble sequence for the three spike trains with different densities, sorted to match the GT indexing and their colors. At low density, the method detects two extra ensembles, denoted by a coloured asterisk. The green vertical region shows an example of ensemble that was correctly detected for the three spike trains. The red vertical region shows an example where the method correctly detects the GT vector for Medium density, but partially fails in the other two cases. **(I)** GT core-cells are sorted in descending order according to the number of core-cells. Each column represents an ensemble, where red indicates a core-cell. **(J-L)** Detected core-cells for the three cases. Despite the two extra ensembles found in **J** (8 and 9, asterisks), core-cells are in good agreement with GT.

**Matching detected ensembles with ground truth ensembles.** To evaluate our method with synthetic data, we matched the GT ensembles with the detected ones. This was implemented by computing the correlation between the GT's activation sequence and the detected ensembles. Thus, we looked for the detected ensemble that maximized the correlation with the GT ensembles.

To test our method, we designed an algorithm to generate synthetic data where neuronal ensembles' activity can be parametrically controlled (see Methods for details). We assessed the detection performance of our method concerning known ground truth (GT).

We illustrate our results with a simple example shown in Fig 2, where different spike trains were generated using a network of $N = 100$ neurons and $T = 5000$ bins (the figure shows just 200 bins to improve the visualization). We created seven ensembles defining the temporal sequence of the ensembles, whose global temporal activity comprised 80% of the sample (Fig 2E), and the core-cells, whose participation comprised from 20 to 40 neurons (Fig 2I). Then, we randomly added or removed spikes to each neuron to satisfy a given firing probability for each neuron, $P(n)$, which controls the spike trains density (Fig 2B–2D).

## Detection of ensembles on synthetic data

We found two ensembles more than expected (seven) for the low-density spike-trains (Fig 2F, red asterisks), while for the medium and high-density ones we found the expected number of ensembles (Fig 2G and 2H, respectively).

Then, we compared the ensemble sequence of the GT (Fig 2E) and the detected ensemble sequence in each density scenario (Fig 2F–2H), finding almost perfect agreement between both, with the exception of a few false positives in the case of low and high density. Finally, we compared the detected core-cells with the GT (Fig 2J–2L), finding good agreement between both. Despite the over-detection in the low-density regime (red asterisk), the other ensembles were in good agreement with the GT core-cells.

With this example, we show that our method can detect the ensemble number, the temporal sequence, and the core-cells of neuronal ensembles in noisy spike trains with different densities. In the next section, we evaluate the detection of three features, i.e., ensemble number, temporal ensemble sequence, and core-cells, for a different sample and network sizes.

## Scaling performance and comparison with an alternative method

To systematically quantify and evaluate the performance of our method (Density-based), we generated synthetic data with different network and sample sizes and compared our algorithm to a current state-of-the-art ensemble detection method published by Carrillo *et al.* (SVD-based) [16]. The associated implementation can be found in https://github.com/hanshuting/SVDEnsemble. We used the default parameters that accompany these codes. Due to the latter method's computational cost, we only compared the scaling with the sample size for both methods, while for the former, we also computed the scaling of performance with network size. The parameters used in our method can be found on the S1 Table. However, they are easily settable in the provided GUI.

First, we generated a synthetic spike train with fixed network size ($N = 300$), number of ensembles ($E = 12$), number of core-cells equal to 35, ensemble probability ($P_E = 0.8$), and medium density. Then, we varied the recording length from $T = 500$ to $T = 10^4$, finding that both methods increased the computational time with the sample size, as expected, but the SVD-based method scaled exponentially, while the Density-based method is two orders of magnitude faster for $T = 10^4$ (Fig 3A).

Regarding the ensemble activity, our method accurately detects the number of ensembles for samples as small as $T = 1000$, while the SVD-based method converges to underestimation of the ensemble number (Fig 3B).

Once the detected ensembles were matched to their closest GT ensemble (see Methods for details), we measured the correlation between both sequences, finding that our method detects the GT with excellent performance over the whole range of sample sizes. The SVD-based method, in turn, systematically fails to detect the global sequence (Fig 3C).

Furthermore, we measured the average correlation between the detected and GT individual sequences, and between the detected and GT core-cells, finding again that our method achieved excellent performance for both features even for small sample sizes (Fig 3D and 3E, respectively).

Finally, to evaluate the performance for different network sizes, we fixed the sample size, while also keeping the other parameters fixed (number of ensembles, of core-cells and $P_E$), and varied the network size from $N = 50$ to $N = 1000$, finding that our method slightly overestimated the ensemble number for small $N$, but yielded accurate results for the larger $N$ (Fig 3F and 3G).

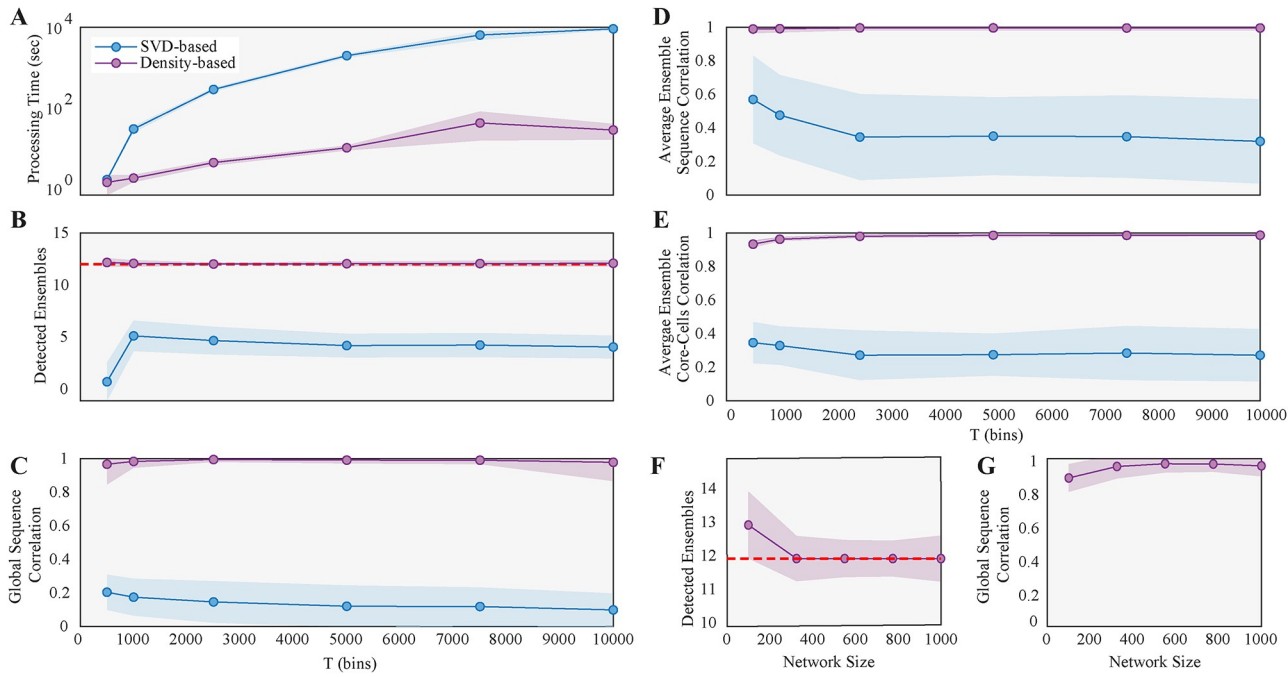

**Fig 3. Performance scaling with sample size and network size. (A)** Computational processing time in log scale of our method (purple) and Carrillo et al [16] (blue), as a function of sample size T (bins). Synthetic data generated using $N = 300$, medium density (see Methods for details) and 35 neurons per ensemble. Dots are averages, and shaded regions are ± 1 standard deviation of 100 repetitions. **(B)** Number of detected ensembles. The dashed red line is the real Number of ensembles. **(C)** Correlation between the detected and true global ensemble sequence. **(D)** Average correlation between detected and true individual ensemble sequence, and **(E)** between detected and true ensemble core-cells. **(F)** Number of detected ensembles as a function of network size. The Number of core-cells per ensemble corresponds to the 35% of the network size. **(G)** Same as **C**, but as a function of network size. Due to computational cost, only our method was evaluated as a function of network size $T = 5000$.

We conclude that our method reliably performs on a wide range of sample and network sizes, giving a more scalable and accurate solution to the ensemble detection problem than the alternative SVD-based method.

## Reliable performance over a wide range of spike-train parameters

Here, we show the performance of our method when the number of ensembles and the number of core-cells vary independently.

We generated synthetic spike-trains with fixed network size ($N = 300$), sample size ($T = 5000$), ensemble probability ($P_E = 0.8$), and medium density while the number of ensembles and core-cells varied, as shown on Fig 4. We found a wide combination of these parameters where the method detects the number of ensembles with a small relative error $(V_O - V_T)/V_T$ where $V_O$ stands for observed value and $V_T$ for theoretical value (Fig 4A), accurately detects the global ensemble sequence (Fig 4B), and the corresponding core-cells (Fig 4C). Further explorations of other parameters and combinations of the same are considered work to be developed. To this end, we provide the computational codes and a GUI at https://github.com/brincolab/NeuralEnsembles. This GUI allows one to perform all the analyses presented here on multi-variate recordings of single events (e.g., spiking data, calcium events, arrival times in a sensor).

## Detection of ensembles on RGC under a simple ON-OFF stimulus

We show our method's usefulness on a single simultaneous recording of a mouse retinal patch *in vitro* using MEA (see Methods for experimental and pre-processing details). We aim to

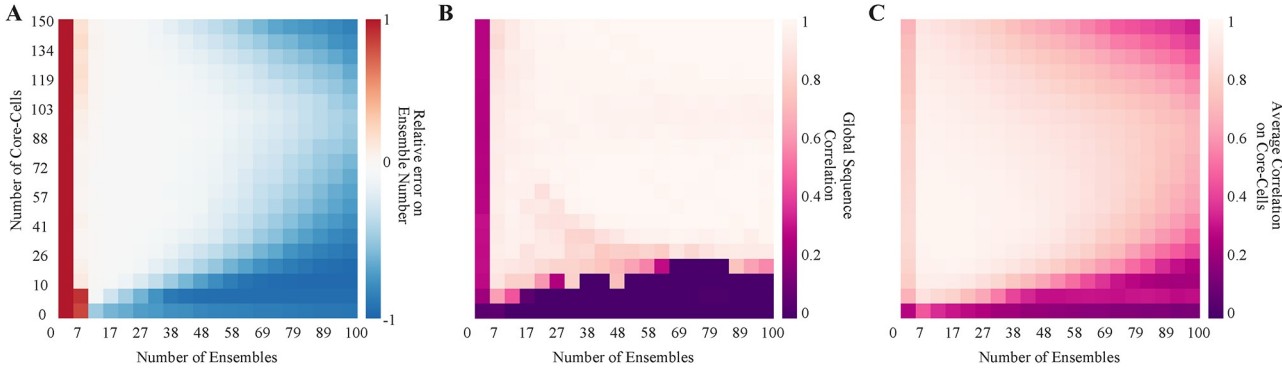

**Fig 4. Detection performance over a wide range of spike-train parameters. (A)** The relative error of detecting the ensemble number for spike-trains with different ensembles and the different number of core-cells. Synthetic data generated using medium density, $N = 300$, and $T = 5000$. Heat maps represent the average over 100 repetitions using the same spike-train parameters. Red/blue represents over/under detection respectively. **(B)** The correlation between the detected and the GT ensemble sequence. **(C)** Average correlation between the detected and the GT core-cells.

illustrate how to interpret the basic results of our method in a particular physiological setting rather than revealing functional properties of the collective activity of the retina. This is the reason why we did not analysed more animals or went deep in the possible implications of retinal neural ensembles for visual processing.

We analyzed the spike response of retinal ganglion cells (RGCs) under a simple ON-OFF light stimulus, where neuronal ensembles are detected without prior information about the stimulus. Their functional role is evaluated in terms of stimulus tuning preference. Data was binarized using a standard bin size of 20ms [45, 46]. We note that our method works on already binarized data, so the binning process is not part of the method. As expected for RGCs, the sum of all the emitted spikes in a given time bin (population activity) is tightly locked to the stimulus, transiently increasing each time the stimulus changes (Fig 5A, top panel). After this harsh response, the population activity decays exponentially until it reaches a stable point. However, the population activity evoked by the ON-stimulus is different in amplitude and shape from the ones evoked by the OFF-stimulus.

These different evoked responses led us to expect at least four types of ensembles: two ensembles related to transitions (one for the ON-OFF and one for the OFF-ON) and two ensembles related to the decaying-stable activity after the ON-OFF and OFF-ON transition, respectively. To test this hypothesis, we applied our ensemble detection algorithm (see Methods for details and parameters) on the spiking activity of 319 RGCs during 120 seconds of MEA recording.

We found ten ensembles comprising ∼68% of the population vectors in the analyzed recording. Their activity was highly locked to the stimulus (Fig 5A middle and bottom panel and D). We found two transiently active ensembles (one for the ON-OFF and one for the OFF-ON transition), whose activity was only evoked by the stimulus transition, showing no activity either before the stimulus start (black arrow in Fig 5A) or during the decaying or stable response. The other eight ensembles were active before stimulus presentation, but at a lower rate, and during the decaying or stable response. Notably, four of them (Ens. 7–10) are preferentially active in the OFF stimulus, and the other four during the ON stimulus (Ens. 2–5).

These results show that the detected retinal ensembles are preferentially tuned to the features of the stimulus, showing that without any stimulus-related information, our method can obtain ensembles whose activity is functionally coupled with the stimulus.

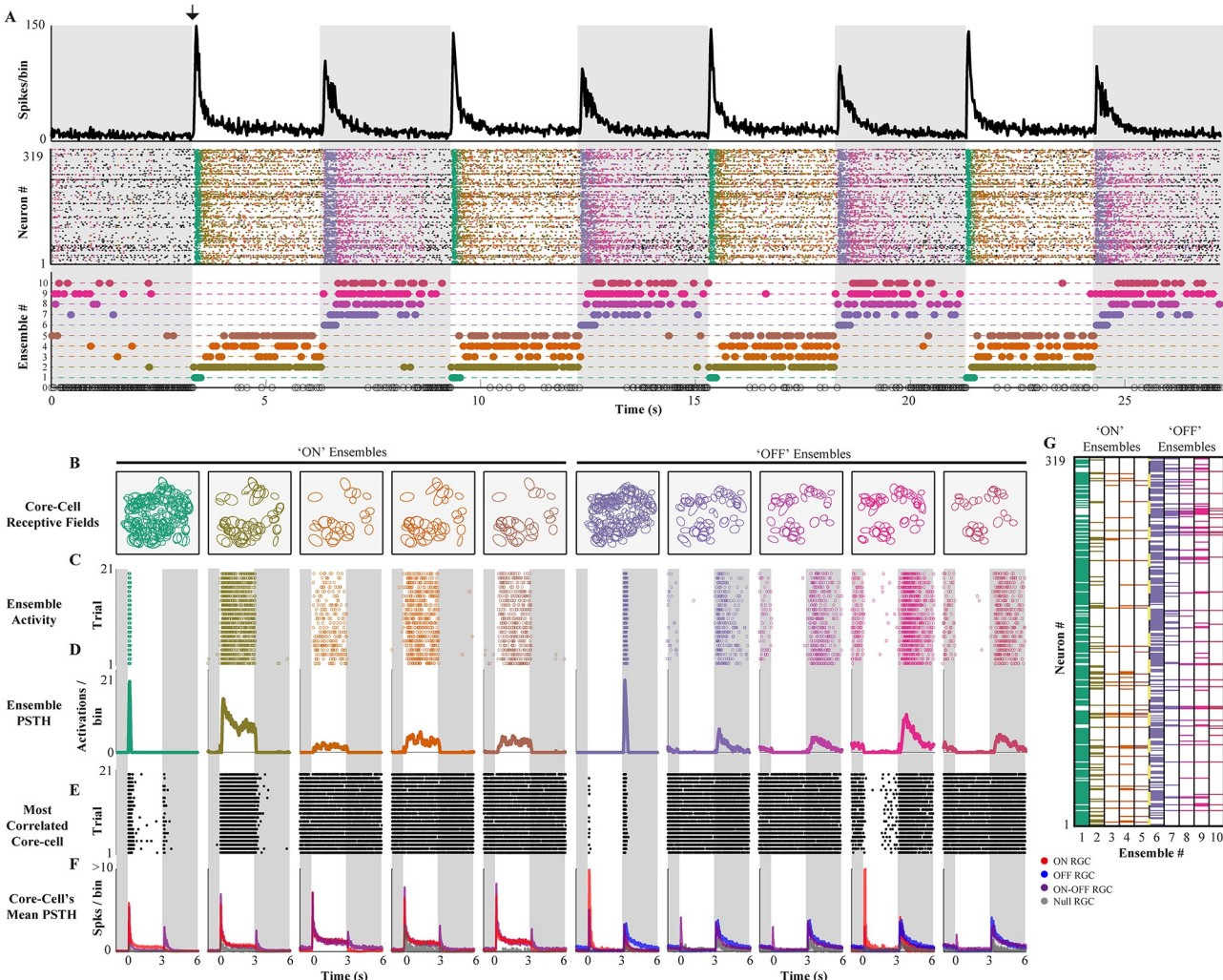

**Fig 5. Stimulus-evoked retinal ensembles. (A) Top.** Population activity (spikes per bin) of 319 RGC's in time during ON light stimulation (white background) and OFF (gray background). The black arrow shows the ON stimulus started. As expected for the retina, the population activity transiently increases when the stimulus switches and then decays and stabilizes. **Middle.** Each population vector is coloured according to the ensemble to which they belong. **Bottom.** The activation of each ensemble in time is represented by a coloured dot (matching the population vector color code). The method detects a consistent stimulus-locked ensemble activity with ON (ensembles from 1 to 5) and OFF (ensembles from 6 to 10) ensembles. Ensemble 0 represents all the population vectors that were discarded according to rejection criteria (see Methods). **(B)** The spatial distribution of the estimated receptive fields for the corresponding core-cells of each ensemble. **(C)** The stacked trial-by-trial responses of each ensemble in time show their reliable and selective responses to specific stimulus features. **(D)** The PSTH for each ensemble shows the stimulus preference of the ON and OFF ensembles. The different responses can be classified as transient and sustained, showing detailed stimulus tuning. **(E)** The stacked trial-by-trial responses of the most correlated core-cell for each ensemble shows that the temporal activity of the ensemble is poorly determined by the activity of their core-cells. **(F)** For each ensemble, the average PSTHs of their corresponding core-cells grouped by cell type (colors) shows no clear tuning preference. **(G)** The coloured matrix shows the RGCs in rows and the ensembles in the columns. Each column is colored in the entries corresponding to their core-cells, according to which ensemble they belong (A).

## Functional properties of RGC ensembles

We detect the ensembles and their *core-cells* i.e. cells whose correlation with a given ensemble is statistically significant(see Methods for details). On average, each core-cell participated in 2.7 ± 1.3 ensembles, and only four RGCs were considered non-core, considering all the ensembles. Three cells participated in six ensembles, indicating that some cells may participate in both ensemble classes.

The two transiently active ensembles, namely Ens. 1 (ON) and Ens. 6 (OFF), have 257 and 222 core-cells, respectively, while the rest of the ensembles have, on average, 47.2 ± 17.9 core-cells (Fig 5G). This result is consistent with the increased population activity evoked by stimulus transition, where many cells are firing, and decaying responses where many cells become silent.

Using the RGC responses to the repeated ON-OFF stimulus (21 trials) and an automated RGC functional classification algorithm (see Methods for details), we obtained 44 ON, 23 OFF, 205 ON-OFF, and 47 Null (no significant preference) RGCs.

All the ensembles were composed of ON-OFF core-cells, but the ON ensembles were dominated by ON RGCs while OFF ensembles by OFF RGCs. Null cells showed significant participation in some ensembles, despite their lack of preference according to the classification algorithm. Indeed, many core-cells participated in many ensembles. For each ensemble, the spatial distribution of the estimated receptive fields (see Methods) of their corresponding core-cells is shown in Fig 5B, revealing this spatial overlap between ensembles. Then, in this simple setup, we found that the ensemble's classification is consistent with their corresponding core-cells' dominant classes.

The detection of core-cells allows us to inquire if the tuning preferences of the detected ensembles are inherited from their core-cells or if the ensembles have a specific tuning preference as a functional unit. We recall that core-cells are not perfectly correlated to the activation times of their corresponding core cells. In other words, the ensemble activation times can not be completely determined by the spike trains of its core-cells, so the tuning preferences of the ensemble as a whole could differ from the tuning preferences of single cells. To qualitatively evaluate this for each ensemble, the peri-stimulus time histogram (PSTH) of each ensemble (Fig 5C and 5D) was computed and compared to the PSTH of their corresponding core-cells (Fig 5E and 5F). Moreover, the most representative core-cell for each ensemble (highest correlation with the ensemble activity, Fig 5E) showed responses that strongly differs from the ensemble responses. Finally, for each ensemble, we averaged the PSTH of the core-cells grouped by RGC class Fig 5F, obtaining one average PSTH per cell class, which also shows the correspondence between ensemble tuning preference and cell type dominance. Since ON-OFF cells are present in all ensembles, the core-cells of each ensemble, as a group, have a preference for both light transitions, despite the precise stimulus tuning of the ensembles (Fig 5D).

As a proof of concept, we test our method on retinal data. We did not went deep on the systematic analysis of retinal ensembles, which we consider a future work to be developed. Nevertheless, our qualitative comparisons suggest that the ensemble tuning preference cannot be completely derived from their core-cells' tuning preference, providing preliminary evidence in favor of neuronal ensembles as whole functional units in the early visual system.

## Discussion

We introduced a scalable and computationally efficient method to detect neuronal ensembles from spiking data. Using simple dimensionality reduction, clustering techniques [43], and suitable statistical tests, we were able to develop a simple, fast and accurate method. On the one hand, our example of mouse retinal ganglion cells provides evidence for an expected causal relationship between stimuli and RGC ensembles. On the other hand, the simulated data examples show that our method provides accurate results for a wide range of neural activity scenarios, outperforming existing tools for ensemble detection in terms of accuracy and computation time.

Our method detects neural ensembles considering four general properties of them: transient activation (spontaneous and/or stimuli-evoked), the presence of a core, the same neurons

may participate in different ensembles, and within-ensemble correlation should be higher than the rest correlation of the whole population. Other methods for ensemble detection fulfill other criteria, e.g., finding communities between spiking neurons along time [26, 47], or using event-related activity [44]. As an alternative to theses these methods, our analysis relies only on grouping the population vectors with no event-related information, allowing us to segment a spiking network's temporal activity under both spontaneous and stimuli-evoked conditions.

Regarding the SVD-based method [9], we acknowledge the insights that its application has provided to the study of neural networks. However, it has limitations in terms of computational cost (for relatively small sample sizes, Fig 3) and parameter tuning. Further, the SVD-based method extracts the temporal activation of ensembles from the spectral decomposition of the similarity matrix between population vectors, aiming to detect groups of linearly independent vectors. Instead, we use a subset of the first principal components to embed the population vectors and cluster them into that space, with no need to find linear independence between the clusters. Finally, the SVD-based method has no explicit implementation for the evaluation of the within-ensemble correlation, which, in our case, is a critical step to distinguish between a noisy cluster and an ensemble cluster.

There is room for further improvement of our method. For example, non-stationarity and non-linear spike correlations may produce spurious results in ensemble detection methods that depend on PCA. Our approach can be adapted to include other measures of spike train similarity besides linear correlation values [48] or include alternative surrogate data methods to provide null hypothesis for correlations that preserve temporal features of spike trains as their inter-spike interval or autocorrelation [49]. However, for generality, we use the least constrained null hypothesis, i.e. to destroy all the spike train temporal structure in order to test the significance of correlation values. For simplicity, in our example of RGCs, we used the standard bin size for mammalian RGCs, but the bin size used to create the spike trains may influence the detected ensembles, and thus different bin sizes can yield different results. Note, however, that our method is designed for already binarized data, so we consider the binning procedure as a pre-processing step, and, as we did, it should be informed by prior knowledge about the temporal resolution of the recorded neural population [45, 46].

Our method does not evaluate the statistical significance of the detected ensemble sequences, which has been proposed to be physiologically meaningful [34]. In fact, the extension toward clustering temporal sequences of population vectors is straight-forward, and we consider its validation and evaluation as a work to be developed. However, our contribution is to put forward a simple, fast, accurate, and scalable algorithm to detect synchronous neural ensembles that, given its simplicity and code availability, can be easily applied in multiple scenarios and extended in several ways.

Finally, we provided novel evidence in favour of the existence of retinal ensembles that are functionally coupled to the stimuli. However, our purpose was to exemplify our method on a biological spiking network rather than explaining the possible mechanisms involved in the activity of retinal ensembles or their functional implications. Indeed, we consider the study of retinal ensembles as an exciting new research avenue that needs to be developed in the way to understand vision, sensory systems, and more generally, the nervous system. In line with this perspective, we delivered a method that is general enough to be applied to any multi-variate binary data set. Furthermore, it is intuitive and can generate results that are easy to visualize, which should favor their comprehension and general use by the scientific community. Matlab codes and a GUI implementing our new method accompany this article.

## Supporting information

**S1 Appendix. Code repository, retinal experiments, and data pre-processing.** Link to the Codes repository and the details of the retinal experiments with their respective pre-processing.
(PDF)

**S1 Table. Method parameters with their description and default values.**
(PDF)

## Acknowledgments

The authors thank Fernando Rosas for insightful discussions and valuable suggestions.

## Author Contributions

**Conceptualization:** Rubén Herzog, Rodrigo Cofré.

**Data curation:** Rubén Herzog.

**Formal analysis:** Rubén Herzog, Rodrigo Cofré.

**Funding acquisition:** Adrian G. Palacios.

**Investigation:** Rubén Herzog, Rodrigo Cofré.

**Methodology:** Rubén Herzog, Soraya Mora, Joaquín Araya.

**Project administration:** Rodrigo Cofré.

**Software:** Rubén Herzog.

**Validation:** Rubén Herzog, Arturo Morales, Soraya Mora.

**Visualization:** Rubén Herzog.

**Writing – original draft:** Rubén Herzog, Rodrigo Cofré.

**Writing – review & editing:** Rubén Herzog, María-José Escobar, Adrian G. Palacios.

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
