## [Decision Letter · Decision Letter 0]

4 Jan 2021

PONE-D-20-33389

Scalable and accurate method for neuronal ensemble detection in spiking neural networks

PLOS ONE

Dear Dr. Herzog,

Thank you for submitting your manuscript to PLOS ONE. After careful consideration, we feel that it has merit but does not fully meet PLOS ONE’s publication criteria as it currently stands. Therefore, we invite you to submit a revised version of the manuscript that addresses the points raised during the review process.

As you will see from the Referees reports, many major points have been raised, and I encourage you to address all of the major points, or explain precisely how they have been taken into account in your revised manuscript (or why they were not).

We look forward to receiving your revised manuscript.

Kind regards,

Jonathan David Touboul

Academic Editor

PLOS ONE

Journal Requirements:

2. Please specify in the methods section how many animals were used for your study.

3.In your Data Availability statement, you have not specified where the minimal data set underlying the results described in your manuscript can be found. PLOS defines a study's minimal data set as the underlying data used to reach the conclusions drawn in the manuscript and any additional data required to replicate the reported study findings in their entirety. All PLOS journals require that the minimal data set be made fully available. For more information about our data policy, please see http://journals.plos.org/plosone/s/data-availability.

Reviewers' comments:

Reviewer's Responses to Questions

**Comments to the Author**

1. Is the manuscript technically sound, and do the data support the conclusions?

Reviewer #1: Yes

Reviewer #2: Partly

Reviewer #3: Yes

2. Has the statistical analysis been performed appropriately and rigorously? 

Reviewer #1: Yes

Reviewer #2: No

Reviewer #3: N/A

3. Have the authors made all data underlying the findings in their manuscript fully available?

Reviewer #1: Yes

Reviewer #2: Yes

Reviewer #3: Yes

4. Is the manuscript presented in an intelligible fashion and written in standard English?

Reviewer #1: Yes

Reviewer #2: Yes

Reviewer #3: Yes

5. Review Comments to the Author

Reviewer #1: In this paper « Scalable and accurate method for neuronal ensemble detection in spiking neural networks », the authors are presenting a new method to identify groups of similarly behaving neurons, and the temporal sequences of these activations. Using both real experimental and synthetic data, they show how a method based on dimensionality reduction combined with a robust density based clustering can identify pattern of activity shared by numerous neurons. Overall, I find the article easy to read and to follow. The methodology is clear and the results well explained. However, I have several concerns before I could recommend the publication of the manuscript.

Major concerns

- My first comment is on the general structure of the manuscript. I think that the experimental validation (with retinal recordings) should come as an illustration of the methods, i.e. should be at the end of the manuscript. The motivation and the scientific questions could remain as they are, but currently, this is a bit odd to start the paper with results performed on real data, before explaining that they are valid because of tests performed on synthetic data. To me, I would switch the two parts, and develop first thoroughly how robust the algorithm is, before applying it to real data.

- As far as I understood, the methods seeks at finding groups of neurons with similar responses. This is particularly useful in the retina, where we know that the retinal ganglion cells have different functional responses, tiling the visual space. However, I found that this information is not used to assess the quality of the ensemble detection with experimental data. Indeed, if the methods succeed at finding groups of neurons, then when applied to the retina, we should recover different subclasses of cells (ON/OFF, sustain vs transient, …). See for example the classification that has been made in [Baden et al, 2016] with calcium recordings. If this is true, then one could look at the tiling of the receptive fields for the neurons in the distinct ensemble, and assess if they are properly covering the space or not. If yes, then this could be a hint than the algorithmic method is finding an ensemble which is likely to reflect a particular subtypes of cells. If no, then it could also be used as an hint than these ensembles are made of more complicated sub-circuits, mixing different subtypes. This is I guess what has been done by the authors in Fig1 panel B, C, D, but I think this would deserve a dedicated figure, as the current take home message is not clear. I would like to see the Receptive Fields, not colored as function of ON/OFF/ON-OFF/Null, but rather as function of their ensemble (as this is currently done for the PSTHs). By doing so, one could appreciate if the ensembles are tiling the space, i.e. if the method is able to distinguish putative cell subtypes.

- Regarding the experimental procedures. How many animals were used to collect the data? When the authors are talking about 319 RGCs during 120s of MEA recordings, is this on a single recording? This comment makes me realize that the tiling mentioned before might not be so obvious, if data are collected across several animals. But if they are, what do the plots in Fig1B corresponds to ? Can the authors explain what these receptive fields maps are, and if data are from a single retina? If, on the other hand, data are from a single retina, why did the authors not reproduce the analysis on various recordings? This would consolidate the (functional) claims on the ensembles, if the number found could be reliable across animals.

- From an experimental point of view, and still in line with was has been done in [Baden et al, 2016], why the authors restricted themselves to very low dimensional stimuli (ON/OFF light stimulus)? If the goal is to detect ensemble, i.e. cells sharing similar responses, then it might be more interesting to display high-dimensional stimuli, to trigger more variability in the responses, and thus get a closer look on these putative ensembles. The more diverse the responses will be, the more the ensemble might appear (assuming there is approximately 30 subtypes of retinal ganglion cells in the mammalian retina, I would naively assume that there should be much more ensembles popping up).

- What is the rationale for discarding responses with low firing rates?

- There is something I do not get with the methods: in the discussion, the authors are talking about the fact that they used “the standard bin size for mammalian RGCs”. I guess this bin size is used to bin the spike trains before PCA, but this is not explicitly said in the methods, is it? Or I may have missed it…Anyway, what is the value used in the paper?

- What is the rationale for allowing only one active ensemble per? I know that otherwise, non-linearity are likely to perturb the methods, but this is a very strong assumption that need to be discussed. While it might be true for retinal activation, assuming different stimuli are triggering different subtypes/responses, one at a time, this assumption does not hold in the global context pictured by the authors in the Introduction. Cortical ensembles in vivo might overlap both in time and space. Please comment on that.

- As the authors acknowledges themselves in the Discussion, different time bins might affect the results. It would be good to provide an analysis on the robustness of the methods w.r.t. the time bin. One could not expect the results to be the same, but a quantification of the variability is important. Since there is no guarantee that the chosen time bin is optimal experimentally, at last it could give some confidence in the results. This can be explored with the synthetic data generated by the authors.

- What about time-warping? I know this might be out of the scope of the manuscript, but maybe it would be worth comparing a bit the method proposed here with the ones seeking at the detection of assemblies taking into account the fact that sequences can be time-warped [Williams et al, 2020, Mackevicius et al, 2019]. In the current paper, the authors are seeking at a fixed sequence of activation between ensembles, but this assumption of a fixed bin size should be discussed.

- What are the equivalent of low/medium/high density cases for real data? More precisely, where are the retina data, on this scale? This should be said to enforce confidence in the results, since the low density case seems to be more problematic.

References

- Williams et al, 2020, Point process models for sequence detection in high-dimensional neural spike trains, https://arxiv.org/abs/2010.04875

- Williams et al, 2020, Discovering Precise Temporal Patterns in LargeScale Neural Recordings through Robust and Interpretable Time Warping, https://doi.org/10.1016/j.neuron.2019.10.020

- Baden et al, 2016, The functional diversity of retinal ganglion cells in the mouse, Nature, https://doi.org/10.1038/nature16468

- Mackevicius et al 2019, Unsupervised discovery of temporal sequences in high-dimensional datasets, with applications to neuroscience https://elifesciences.org/articles/38471

Reviewer #2: In this manuscript the authors introduce a new pipeline for the detection of synchronous cell assemblies in parallel spike trains. The proposed algorithm relies on density-based clustering of population activity patterns. The algorithm performance is assessed on simulated ground truth and then tested on spike trains from in vitro retinal ganglion cells during an ON-OFF light stimulation.

I found the simplicity and the scalability (huge problem for the community) of the algorithm the strength points of the proposed manuscript, which I found useful for the scientific community. However, there are a number of major points to address before publication.

Major points:

1. Assessment of core-cells: Once established the assembly core patterns (centroids) the authors define the core-cells of each assembly by testing the correlation between single-unit activity and assembly signals. This is done by shuffling the temporal sequence of assembly activations. While I found appropriate using bootstrap techniques to assess the correlation significance, in case of time series it is important to preserve the internal autocorrelation of the shuffled signal to avoid false discoveries. The most common, but not exclusive, way of doing so is by first dividing the time series in windows of size comparable to the signal autocorrelation length, and then by permuting the order of such windows [Efron & Tibshirani, Introduction to the bootstrap, 1993, Springer]. I, therefore, suggest the authors to address this point and modify the algorithm accordingly. Also, the authors leave the significance parameter of this test as free parameter. This should not be the case, as significance it is not arbitrary, and this parameter should be limited to <0.05.

2. The simulated time series used to test the validity of the method are quite unrepresentative of real biological data: a) they are stationary; b) the ratio between assembly activations and background spikes is extremely high, to the point that (in the low and medium density case) almost all spikes fired by a unit take part to one or the other assembly; c) as mentioned in page 11 lines 295-296, non-assemblies spike patterns are filled with only silent spikes. In these extreme conditions is a bit difficult to assess the algorithm qualities. I would therefore suggest testing the false discovery rate of the method by using a nonstationary background (non-assembly) activity for all units, and a way smaller ratio between assembly and non-assembly activity. In particular, I suggest formally evaluating the false discovery rate (in terms of detected core-units and assemblies) for a non-stationary background activity and: 1) a range of values of total assembly activations; 2) a range of values of percentages of assembly spikes missing (similar in spirit to what done for Fig 3B).

3. I imagine that detecting assemblies composed of very few core-units might be challenging for this kind of approach. I would therefore suggest discussing this limitation (if present).

4. As last step of the algorithm “The inner-cluster mean correlation is compared against a threshold of non-core cell correlation, discarding any cluster below this threshold”. This final step is not clearly explained and would benefit of some more detail. How is this threshold chosen?

5. The sentence “Among them, only the one based on the correlation between population patterns, and the one based on non-negative matrix factorization, fit with a definition of ensembles.” is either false or confusing. I would suggest rephrasing or removing.

6. In the discussion, the authors compare their method with other state of the art techniques (Page 13, lines 372-382). I found this paragraph a bit surprising, as a large variety (I would say the majority) of assembly detection techniques can detect synchronous assembly patterns and with no-event related information. I would therefore suggest to rephrase and tone down this paragraph and, maybe, focus more on the method scalability, real point of strength.

Minor points:

7. Please, define mathematically the quantity “relative error”.

8. Please, indicate the value of all arbitrary parameters used for each analysis (also for the SDV-based method).

9. There are relatively many parameters that the authors leave free for the user to decide. While user necessities are multiple and have to adapt to diverse datasets, a total arbitrariness of these choices might lead to extreme settings and unreasonable assembly detections. I think it would beneficial for future users to discuss the implications behind each parameter choice (e.g. by showing the consequences of choosing a different number of PCA principal components when many assemblies are present) and to suggest a parameters set “safe” for the users to use.

Reviewer #3: Comments to manuscript: “Scalable and accurate method for neuronal ensemble detection in spiking neural networks” PONE-D-20-33389

This manuscript proposes an analytical method to detect neuronal ensembles from MEA electrical recordings. The work demonstrates that the method could be used in retinal recordings exposed to ON-OFF light stimuli. The authors validated the method using synthetic data to identify different neuronal ensembles. The standardization of methodologies to identify groups of neurons with coordinated activity that are related to specific experimental conditions is a relevant topic in neuroscience since the simultaneous recordings of neuronal populations has become worldwide available and is growing at a steady pace. The manuscript describes a potential method to identify neuronal ensembles, but several concepts and implementations should be clarified in order to make the method useful for diverse brain structures. Otherwise, the tittle should be changed to highlight that the method is useful mainly for in vitro retinal recordings and specific experimental conditions.

Finally, the analysis for the sequential organization of the ensembles needs to demonstrate that sequential patterns of activity could not be identified in random population activity.

Major:

1) The definition of what is the meaning of a neuronal ensemble is not clear from the abstract nor the introduction. This is a relevant issue since the goal of the manuscript is to detect neuronal ensembles.

It is mentioned that a neuronal ensemble is a group of neurons that fire together (page 1, line 9) but the authors indistinctively call a neuronal ensemble a group of neurons firing together or different groups of neurons firing in sequences. The methodological demonstration of groups of neurons firing together or groups of neurons firing in sequences are two completely different approaches.

2) Page 2, line 42. What is the meaning of core cells? Does core cells represent the same concept of a neuronal ensemble? In such case the use of core is unnecessary and confusing.

3) Page 3, lines 72-79. The use of the concept of core-cells is not clear. Why “the ensembles as a whole are not a simple inheritance from their corresponding core-cells” if the authors define a neuronal ensemble as the core-cells in the previous page.

4) Page 3, line 83. It is not clear what is the meaning of scalable. Does it mean different brain areas or different orders of magnitude in the number of recorded neurons? Or different lengths and sizes of the data?

5) Page 3, lines 95-96. In the example shown in figure 1A, it is clear that ON and OFF early responses have different properties as length and amplitude. In particular, the approach used in the manuscript (PCA) will enhance such differences in the low dimensional projection of the data. However, in many other conditions and brain structures the activity of different ensembles could have the same properties limiting the use of PCA to separate such subtle changes. The authors should show that ensembles with the same overall characteristics of population activity could be detected with PCA. Otherwise, the manuscript should clarify that this method is useful for retinal recordings.

6) Figure 1A. From the figure Bottom it is hard to see if many ensembles could be active at the same time? If this is the case, then the definition of a neuronal ensemble becomes critical for the claims of the manuscript. Because, in that case several combinations of different groups could potentially represent different neuronal ensembles.

6) Figure 1B. The labeling is confusing because it is mentioned before that four different ensembles are expected. What is the meaning of ON-OFF?

7) Page 5, lines 148-150. It is not clear what measurement was used to reach the conclusion that the ensemble tuning preference cannot be completely derived from their core-cells.

8) Even though the method is intended for a broad neuroscience community. It is important to mention in the results how the spikes were extracted and how the ON and OFF cells could be identified.

9) Page 5, line 156. It should be indicated in the results what is the meaning of a fixed number of spikes.

10) Page 5, line 159. It is mentioned that only 3 to 6 principal components are used. The authors should indicate in the results section what is the criteria to select 3, 4, 5 or 6 principal components.

11) Page 8. Section Detection of ensembles on synthetic data. To test the method the authors generated synthetic data that resembles the data showed previously from retina. As I mentioned before the manuscript shows that the method is ad hoc for retina recordings but not other brain areas.

12) Page 8, lines 195-206. The manuscript introduced here the detection of sequential activity patters. But the criteria to define sequential patterns of activity and their statistical significance were not mentioned anywhere. Such criteria should be included otherwise should be removed from the paper.

13) Pages 8-9. To evaluate and compare the performance of the method the manuscript compared the results obtained with synthetic data using the current method and a previously published method (SVD-based). As I mentioned before the authors showed that their method is useful for data with specific characteristics so it is expected that the performance will excel other methods. On the other hand, the method compared (SVD-based) was used for optical recordings whereas the current method analyzed electrical recordings. The difference in the temporal resolution of the data could also bias the results. The authors should compare their method against other method applied to electrical recordings or discuss the limitations to apply it to data obtained with lower temporal resolution.

14) Page 10. Synthetic Spike Trains section. It is not clear why the authors used a Bernoulli process to define the activation sequence of each ensemble. A Bernoulli process define a series of random events that are independent whereas the sequential activity of neuronal ensembles has been shown to be time dependent.

15) Page 11. Feature extraction using PCA section. The authors should be consistent in the terminology they used along the manuscript. This section mentions that spike patterns with less than 3 spikes were discarded. What is a spike pattern? Is this a population pattern or a single cell pattern? Why 3 spikes?

16) Page 11. Previously the manuscript mentioned that 3 to 6 principal components (page 5, line 159) were used but now they mention that 4 to 7 principal components were used (page 11, line 309).

17) Page 11, line 313. What is close pattern’ clustering?

18) Page 12, Centroid detection section. It is not clear from the text how the clustering is performed. Is it a hard or a soft clustering?

19) Did the authors remove the points that couldn’t be clustered in the low dimensional space and then perform the clustering again? The main limitation of clustering high dimensional data is that all the points will appear in the low dimensional representation even though they represent unique events.

20) What is a closest centroid? The manuscript didn’t mention any measurement for that.

21) Page 12. Core-cell detection section. This section described the procedure to detect core-cells. However, from the text it is not clear why these core-cells are important or what is the difference between an ensemble and the core-cells of an ensemble. Is it the same?

22) Page 12. Ensemble selection criteria. The authors mention two criteria for ensemble selection. But from the text it is not clear how to choose such parameters. This is important for anyone interested in applying the method to their own data.

23) Page 13, line 386. The SVD-based method from the reference cited (16) didn’t extract a temporal sequence of ensembles as mentioned in the text.

24) The authors should compare different bin sizes as they mentioned in the discussion. This is very important for the broad use of the method.

Minor:

1) Page 2, line 39. It is not clear from the text what is the meaning of “one-time bin of the spike train”.

2) Page 3, lines 66-67. Revise “…when confronted to changes light intensity”

3) Page 3, line 88. What is parallel recording? Does it mean simultaneous?

4) Page 4, lines 114-115. The conclusion of this sentence is not clear since the data only shows ON and OFF responses. What does it mean without any stimulus-related information?

6. PLOS authors have the option to publish the peer review history of their article (what does this mean?). If published, this will include your full peer review and any attached files.

Reviewer #1: No

Reviewer #2: No

Reviewer #3: **Yes: **Luis Carrillo-Reid

---

## [Author Response · Author response to Decision Letter 0]

4 Feb 2021

We opt to make a pdf file with the response to the reviewers. We attached this pdf as Reviewer Response in the 'Attach Files' section.

---

## [Decision Letter · Decision Letter 1]

1 Apr 2021

PONE-D-20-33389R1

Scalable and accurate method for neuronal ensemble detection in spiking neural networks

PLOS ONE

Dear Dr. Herzog,

Thank you for submitting your manuscript to PLOS ONE. After careful consideration, we feel that it has merit but does not fully meet PLOS ONE’s publication criteria as it currently stands. Therefore, we invite you to submit a revised version of the manuscript that addresses the points raised during the review process.

As you will see, Reviewers 1 and 3 are satisfied with the work, while Reviewer 2 has repeated or made more specific some of their initial requests. Given the feedback of the Referees and the nature of the additional requests, it seems that the paper will likely meet the publication criterial of Plos One. However, I wanted to give you an opportunity to access this report and, when you judge it appropriate, alter your manuscript accordingly; for points you do not wish to pursue, please provide a brief rebuttal in your resubmission letter.

We look forward to receiving your revised manuscript.

Kind regards,

Jonathan David Touboul

Academic Editor

PLOS ONE

Journal Requirements:

Reviewers' comments:

Reviewer's Responses to Questions

**Comments to the Author**

1. If the authors have adequately addressed your comments raised in a previous round of review and you feel that this manuscript is now acceptable for publication, you may indicate that here to bypass the “Comments to the Author” section, enter your conflict of interest statement in the “Confidential to Editor” section, and submit your "Accept" recommendation.

Reviewer #1: All comments have been addressed

Reviewer #2: (No Response)

Reviewer #3: All comments have been addressed

2. Is the manuscript technically sound, and do the data support the conclusions?

Reviewer #1: Yes

Reviewer #2: Partly

Reviewer #3: Yes

3. Has the statistical analysis been performed appropriately and rigorously? 

Reviewer #1: Yes

Reviewer #2: No

Reviewer #3: N/A

4. Have the authors made all data underlying the findings in their manuscript fully available?

Reviewer #1: Yes

Reviewer #2: Yes

Reviewer #3: Yes

5. Is the manuscript presented in an intelligible fashion and written in standard English?

Reviewer #1: Yes

Reviewer #2: Yes

Reviewer #3: Yes

6. Review Comments to the Author

Reviewer #1: In this revised manuscript, most of my comments/concerns have been addressed, and I can now recommend its publication

Reviewer #2: To my major surprise, the Authors decided not to address the majority of the points raised as major concerns. Without an appropriate reply to all points raised in my original review (and with this I mean the addition in the manuscript of new tests and a discussion of the algorithm limitations) at the next round of revision I will be forced to reject the paper.

Specifically, in relation to the Author’s replies:

1) First, as visible from Fig.5A retinal ganglion cells have a clear autocorrelation. Secondly, and most importantly, the goal of the paper is to provide the scientific community with a tool to detect cell assemblies in experimental data. It is well established that neuronal time series are non-stationary and autocorrelated, ignoring this fact leads to a wrong significance assessment and false discovery rates. The fact that the synthetic data used in the paper are stationary just means that they were not suitable to test the algorithm (and this is why in point 2 of my original review I asked the Authors to test the algorithm on more realistic time series). I am very surprised that the Authors decided to neglect this fundamental request, both because of its major importance and because it can be easily fixed (as explained in point 1 of my original review).

As side note, a significance value above 0.05 means that the result is non-significant. Increasing the alpha value (that is the threshold for significance) can not solve problems of low statistical power.

2) The purpose of the synthetic data is exactly to reproduce all characteristics of the biological data which might affect the reliability of the algorithm. The algorithm is presented as a general methodology to test cell assemblies and not to test exclusively retinal ganglion cells (if it was explicitly aimed to test retinal ganglion cells, this should be clearly stated and the Authors would still have to change their synthetic data, since the data in Fig. 2 don’t have the same characteristic of autocorrelation of those shown in Fig. 5A). Of course I understand that all algorithms have some limitations, but such limitations have to be tested and addressed in the manuscript to inform future users. In this manuscript this assessment is very poor and misleading.

3) Again, the limitations of the method have to be tested and described. I might have missed something but to my understanding it has not been studied how the assembly discovery rate varies as function of the assembly core size. Clarifying this point will be of great relevance for future users who might interpret why they can not find assemblies if the number of core cells is much smaller of the whole population number (if this is true, it should be indeed be tested).

4) Thanks for the clarification. Since the threshold for ensemble selection comes from the average pairwise correlation of the whole population, would this create a problem if e.g. all units of the tested sample take part in one global-assembly? Would this whole population-assembly be detected?

6) I had understood that the presented method does not rely on any event related information. Point 6 of my old review referred to the Author’s sentence “Despite the usefulness of these methods in their context, our analysis is more general. It relies on grouping the population spiking patterns with no event-related information …” This sentence hint at the fact that the majority of the available methods use event-related information. This is not true, since the majority of cell assembly detection methods indeed do not. I suggest correcting this sentence accordingly.

8) All the free parameters used should be available to the reader without forcing him to dig into the code.

Reviewer #3: The authors followed all my suggestions and answer my concerns, including their changes in the new version of the manuscript.

7. PLOS authors have the option to publish the peer review history of their article (what does this mean?). If published, this will include your full peer review and any attached files.

Reviewer #1: No

Reviewer #2: No

Reviewer #3: **Yes: **Luis Carrillo-Reid.

---

## [Author Response · Author response to Decision Letter 1]

28 Apr 2021

The responses to the reviewers is attached in a pdf file.

---

## [Editor Report · Decision Letter 2]

30 Apr 2021

Scalable and accurate method for neuronal ensemble detection in spiking neural networks

PONE-D-20-33389R2

Dear Dr. Herzog,

We’re pleased to inform you that your manuscript has been judged scientifically suitable for publication and will be formally accepted for publication once it meets all outstanding technical requirements.

Kind regards,

Jonathan David Touboul

Academic Editor

PLOS ONE
---

## [Editor Report · Acceptance letter]

22 Jul 2021

PONE-D-20-33389R2 

Scalable and accurate method for neuronal ensemble detection in spiking neural networks  

Dear Dr. Herzog:

I'm pleased to inform you that your manuscript has been deemed suitable for publication in PLOS ONE. Congratulations! Your manuscript is now with our production department. 

Kind regards, 

on behalf of

Dr. Jonathan David Touboul 

Academic Editor

PLOS ONE